# Proteomic profiling of the local and systemic immune response to pediatric respiratory viral infections

Emily Lydon,[1] Christina M. Osborne,[2,3] Brandie D. Wagner,[4] Lilliam Ambroggio,[5,6] J. Kirk Harris,[6] Ron Reeder,[7] Todd C. Carpenter,[6] Aline B. Maddux,[6] Matthew K. Leroue,[6] Nadir Yehya,[2] Joseph L. DeRisi,[8,9] Mark W. Hall,[10] Athena F. Zuppa,[2] Joseph Carcillo,[11] Kathleen Meert,[12] Anil Sapru,[13] Murray M. Pollack,[14] Patrick McQuillen,[15] Daniel A. Notterman,[16] Charles R. Langelier,[1,8] Peter M. Mourani,[17] for the Eunice Kennedy Shriver National Institute of Child Health and Human Development Collaborative Pediatric Critical Care Research Network (CPCCRN)

ABSTRACT  Viral lower respiratory tract infection (vLRTI) is a leading cause of hospitalization and death in children worldwide. Despite this, no studies have employed proteomics to characterize host immune responses to severe pediatric vLRTI in both the lower airway and systemic circulation. To address this gap, gain insights into vLRTI pathophysiology, and test a novel diagnostic approach, we assayed 1,305 proteins in tracheal aspirate (TA) and plasma from 62 critically ill children using SomaScan. We performed differential expression (DE) and pathway analyses comparing vLRTI ($n$ = 40) to controls with non-infectious acute respiratory failure ($n$ = 22), developed a diagnostic classifier using LASSO regression, and analyzed matched TA and plasma samples. We further investigated the impact of viral load and bacterial coinfection on the proteome. The TA signature of vLRTI was characterized by 200 DE proteins ($P_{adj}$ <0.05) with upregulation of interferons and T cell responses and downregulation of inflammation-modulating proteins including FABP and MIP-5. A nine-protein TA classifier achieved an area under the receiver operator curve (AUC) of 0.96 (95% CI: 0.90–1.00) for identifying vLRTI. In plasma, the host response to vLRTI was more muted with 56 DE proteins. Correlation between TA and plasma was limited, although ISG15 was elevated in both compartments. In bacterial coinfection, we observed increases in the TNF-stimulated protein TSG-6, as well as CRP, and interferon-related proteins. Viral load correlated positively with interferon signaling and negatively with neutrophil-activation pathways. Taken together, our study provides fresh insights into the lower airway and systemic proteome of severe pediatric vLRTI and identifies novel protein biomarkers with diagnostic potential.

IMPORTANCE  We describe the first proteomic profiling of the lower airway and blood in critically ill children with severe viral lower respiratory tract infection (vLRTI). From tracheal aspirate (TA), we defined a proteomic signature of vLRTI characterized by increased expression of interferon signaling proteins and decreased expression of proteins involved in immune modulation including FABP and MIP-5. Using machine learning, we developed a parsimonious diagnostic classifier that distinguished vLRTI from non-infectious respiratory failure with high accuracy. Comparative analysis of paired TA and plasma specimens demonstrated limited concordance, although the interferon-stimulated protein ISG15 was significantly upregulated with vLRTI in both compartments. We further identified TSG-6 and CRP as airway biomarkers of bacterial-viral coinfection, and viral load analyses demonstrated a positive correlation with interferon-related protein expression and a negative correlation with the expression of neutrophil activation proteins. Taken together, our study provides new insights into the lower airway and systemic proteome of severe pediatric vLRTI.

Editor Neha Garg, Georgia Institute of Technology, Atlanta, Georgia, USA

Address correspondence to Charles R. Langelier, chaz.langelier@ucsf.edu.

Emily Lydon and Christina M. Osborne contributed equally to this article. The author's order was determined alphabetically.

Charles R. Langelier and Peter M. Mourani are joint senior authors.

The authors declare no conflict of interest.

See the funding table on p. 11.

KEYWORDS   proteomics, LRTI, pneumonia, host response, co-infection, viral pneumonia, respiratory viruses

Respiratory viral infections are the most common cause of pediatric illness worldwide (1). Although often mild and self-limited, a substantial number of children progress to severe viral lower respiratory tract infection (vLRTI) requiring hospital admission and mechanical ventilation (MV), often further complicated by acute respiratory distress syndrome (ARDS) and/or bacterial coinfections. In a global epidemiological study of children under five, severe LRTI was the leading cause of mortality outside of the neonatal period, contributing to an estimated 760,000 deaths (2).

The marked heterogeneity in vLRTI clinical outcomes, driven in large part by differential host responses, remains poorly understood (3). Deeply profiling the host immune response to vLRTI can offer insights into pathophysiology and also enable novel diagnostic test development (4, 5). Prior work evaluating the host response in LRTI using systems biology approaches has mainly focused on adult populations, and the few pediatric LRTI studies predominantly utilized transcriptomic or metabolomic approaches (6–10). Proteomics, or the large-scale study of the protein composition within a biologic sample, has the potential to complement studies of the transcriptome, as protein expression is influenced by post-transcriptional regulation and may be a more direct reflection of cellular and immunologic processes (11). The limited number of proteomic pediatric LRTI studies published to date have profiled plasma or urine samples (12–14), which provide useful insights into the systemic response to LRTI and offer candidate diagnostic biomarkers but may not reflect biological processes at the site of active infection. The local host proteomic response to severe viral infection in the lower respiratory tract remains poorly understood in children, as does the compartmentalization of proteomic responses in the blood versus airway.

To address these questions, we perform high-dimensional proteomic profiling of paired tracheal aspirate (TA) and plasma samples in a prospective multicenter cohort of critically ill children with acute respiratory failure, specifically comparing vLRTI to non-infectious etiologies. We hypothesized that there would be a distinct proteomic signature of vLRTI, more pronounced in the airway than blood, and that exploring proteomic correlations with bacterial-viral coinfection and viral load would yield valuable biological insights.

## MATERIALS AND METHODS

### Description of cohort

Children in this study represent a subset of those enrolled in a previously described prospective cohort of 454 mechanically ventilated children admitted to eight pediatric intensive care units in the National Institute of Child Health and Human Development's (NICHD) Collaborative Pediatric Critical Care Research Network (CPCCRN) from February 2015 to December 2017 (8, 9). See supplementary material for enrollment criteria. IRB approval was granted for TA sample collection prior to consent, as endotracheal suctioning is standard-of-care. Specimens of children for whom consent was not obtained were destroyed. The study was approved by University of Utah IRB #00088656.

### Sample collection and processing

TA specimens collected within 24 h of intubation were processed for proteomic analysis, with centrifugation at 4°C at 15,000 × $g$ for 5 min and freezing of supernatant at −80°C in a microvial within 30 min. Some patients did not have TA samples available for proteomic analysis due to inadequate processing. Plasma samples collected within 24 h of MV were frozen at −80°C. Some patients did not have plasma collected because consent was not obtained within the timeframe.

## Adjudication of LRTI status

Adjudication was carried out retrospectively by study-site physicians who reviewed all clinical, laboratory, and imaging data following hospital discharge, with specific criteria detailed in the supplementary material. Standard-of-care microbiological testing, including multiplex respiratory pathogen polymerase chain reaction (PCR) and semiquantitative bacterial respiratory cultures, was considered in the adjudication process. In addition, microbes detected by TA metagenomic next-generation sequencing (mNGS), as previously described (8), were considered for pathogen identification. Patients were assigned a diagnosis of "vLRTI" if clinicians made a diagnosis of LRTI, and the patient had a respiratory virus detected by PCR and/or mNGS. Within the vLRTI group, subjects were subcategorized as viral infection alone or bacterial coinfection, based on whether a bacterial respiratory pathogen was detected by bacterial culture, PCR, and/or mNGS. Patients alternatively were assigned a diagnosis of "No LRTI" if clinicians identified a clear, non-infectious cause of respiratory failure without clinical or microbiologic evidence of bacterial or viral LRTI.

## Subject selection for proteomic analysis

Subjects that were clinically adjudicated as vLRTI and No LRTI were selected for proteomics analysis, in an approximately 2:1 ratio. This subset of subjects represented a convenience sample of the larger cohort, with the goal of maximizing the number of subjects with both TA and plasma samples to allow comparative proteomic analysis, although not all subjects had all both samples available.

## Proteomic analysis

The SomaScan 1.3 k assay (SomaLogic) was utilized to quantify the protein expression in plasma and TA samples. The assay, described and validated elsewhere (15–17), utilizes 1,305 single-stranded DNA aptamers that bind specific proteins, which are quantified on a customized Agilent hybridization assay. Aptamer measurement is therefore a surrogate of protein expression. The assay outputs fluorescence units that are relative but quantitatively proportional to the protein concentration in the sample.

## Statistical analysis

Relative fluorescence units (RFUs) for each of the 1,305 protein aptamers were log-transformed for analysis. Differential expression was calculated between groups for each aptamer using limma, a R package that facilitates simultaneous comparisons between numerous targets (18). Age-adjusted and age-unadjusted differential protein analyses were performed. Biological pathways were interrogated against the Reactome database with the R package WebGestaltR using a functional class scoring approach (19, 20). Specifically, the input list included the full set of 1,305 proteins and the corresponding log2-fold change between the conditions of interest, ranked by T-statistic. *P* values for protein and pathway analyses were adjusted for multiple comparisons using the Benjamini-Hochberg procedure; adjusted *P* value ($p_{adj}$) <0.05 was considered statistically significant.

A parsimonious proteomic classifier was generated using LASSO logistic regression on TA samples with the cv.glmnet function in R, setting family = "binomial" and leaving other parameters as default (21). LASSO was used for both feature selection and classification. The model was generated using 5-fold cross-validation, where a model was trained on ~80% of samples and tested on ~20% of samples to generate vLRTI probabilities for each of the subjects in the cohort. To keep the fold composition comparable, we required at least 3 No LRTI subjects in each fold. Area under the receiver operator curve (AUC) was calculated using the pROC package, and confidence intervals were generated with bootstrapping (22).

Correlation for each protein between TA and plasma samples was calculated using Pearson correlation for all paired samples in bulk and then subdivided by group (vLRTI vs

No LRTI). Correlation coefficients were considered strong if the absolute value was >0.5, moderate if 0.3–0.5, and weak if 0–0.3. Correlation between specific proteins and viral load was calculated similarly. Viral load was extrapolated from mNGS reads-per-million, and if multiple viruses were detected, the viral loads were summed.

## RESULTS

### Cohort characteristics and microbiology

From the prospective multi-center cohort ($n = 454$), samples from 62 subjects underwent proteomic analysis, including 40 with vLRTI and 22 with No LRTI (Fig. 1A). Those with vLRTI were further subdivided into viral infection alone ($n = 16$) or viral-bacterial coinfection ($n = 24$). The demographic characteristics did not differ between the vLRTI and No LRTI groups, with the exception of age, which was higher in No LRTI than vLRTI (median 10.2 years [IQR 1.1–14.9] vs 0.9 [0.3–1.6]) (Table 1). Diagnoses in the No LRTI group included trauma, neurologic conditions, ingestion, and anatomic airway abnormalities, with many having abnormal chest radiographs and meeting ARDS criteria. Within the vLRTI group, respiratory syncytial virus (RSV) was the most common pathogen, and 15 subjects had more than one virus (Fig. 1B). *Haemophilus influenzae*, *Moraxella catarrhalis*, and *Streptococcus pneumoniae* were the most common coinfecting bacterial pathogens.

### Defining a lower respiratory tract proteomic signature of vLRTI

We first compared protein expression in TA samples between the vLRTI ($n = 37$) and No LRTI ($n = 18$) groups. Two hundred proteins (15.3% of all proteins assayed) were differentially expressed at $p_{adj} < 0.05$ (Fig. 2A and B). Among the 80 proteins upregulated in vLRTI were interferon-stimulated ubiquitin-like protein ISG15 and oligoadenylate synthase protein OAS1, which are central to type I interferon signaling, and Granzyme B and Granulysin, proteins present in granules of cytotoxic T cells and natural killer (NK) cells. Among the 120 proteins downregulated in vLRTI were fatty acid-binding protein FABP, macrophage inhibitory protein MIP-5, and neutrophil-activating protein NAP-2. Pathway analysis confirmed interferon signaling as the primary pathway upregulated in the vLRTI group, although only the "influenza infection" pathway achieved $p_{adj} < 0.05$ (Fig. 2C).

Having identified a strong host proteomic signature of vLRTI, we hypothesized that a parsimonious number of TA proteins could accurately differentiate vLRTI from No LRTI subjects. Utilizing LASSO logistic regression and employing five-fold cross-validation, we built parsimonious proteomic classifiers (ranging in size from 9 to 15 proteins) that accurately distinguished vLRTI and No LRTI with an AUC of 0.96 (95% CI: 0.90–1.00) (Fig. 2D; Table S1). The proteins with consistently positive coefficients (i.e., increasing vLRTI probability) were Granulysin, Granzyme B, and ISG-15 as well as cyclin-dependent kinase protein CDK2 and kinesin-like protein KIF23. The proteins with consistently negative coefficients (i.e., decreasing probability of vLRTI) were FABP and NAP-2.

Since age was statistically different between the two groups, we added age as a continuous covariate in our differential expression model (Fig. S1). The results overall were similar, with 176 differentially expressed proteins (58 upregulated and 118 downregulated with vLRTI). There was considerable overlap (80%) in the top 10 most differentially expressed proteins between the two models.

### Comparison of plasma proteomics between vLRTI and No LRTI groups

We next compared plasma protein concentrations between vLRTI ($n = 33$) and No LRTI ($n = 22$) groups. The age-unadjusted differential expression analysis yielded 56 statistically significant proteins (4.3% of all proteins assayed)–45 upregulated in vLRTI and 11 downregulated in vLRTI (Fig. 3A). However, adjusting for age, only one protein, ISG15, remained significant (Fig. S2). ISG15, a type 1 interferon-stimulated protein, showed promise in distinguishing vLRTI and No LRTI groups in both plasma and TA ($p_{adj}$ for

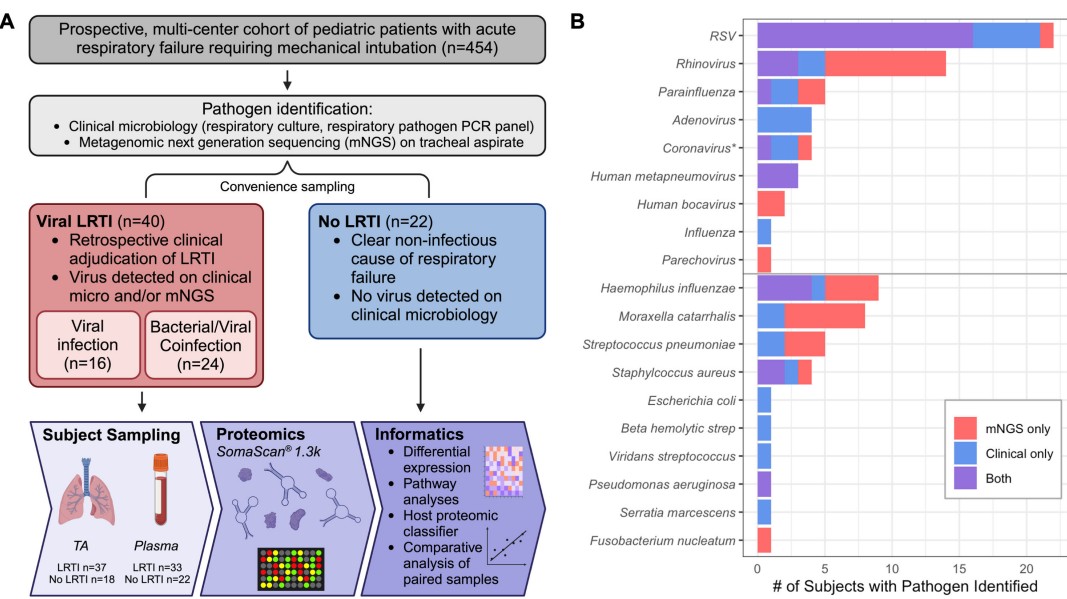

**FIG 1** Study overview. (A) From a prospectively enrolled multicenter cohort of pediatric patients presenting with acute respiratory failure requiring intubation, 62 were selected for proteomic analysis. The vLRTI group included a subset of subjects who had bacterial coinfection. Plasma and tracheal aspirate (TA) samples collected on enrollment underwent proteomic profiling on the SomaScan platform. Some subjects did not have all samples available for analysis; hence, the numbers available for each sample type are shown. Informatics approaches included the evaluation of differentially expressed proteins and pathways, development of a host proteomic classifier, and comparative analyses of paired samples. (B) Microbiology of the vLRTI group. Bar plot color indicates whether the microbe was detected on clinical microbiology, tracheal aspirate metagenomic next-generation sequencing (mNGS), or both. Many subjects had co-detection of multiple pathogens; thus, the total number of pathogens exceeds the number of patients in the vLRTI cohort. *Coronavirus includes only non-SARS-CoV-2 coronaviruses.

both <0.0001), and ISG15 expression was strongly correlated between paired TA and plasma samples (correlation coefficient 0.79, $P < 0.0001$) (Fig. 3B and C). ISG-15 alone implemented as a diagnostic test exhibited strong performance with AUCs of 0.95 (95% CI: 0.89–1.00) and 0.91 (95% CI: 0.83–0.99) in TA and plasma, respectively (Fig. S3).

## Comparative analysis of plasma and respiratory tract proteomics

Comparing the differentially expressed proteins between vLRTI and No LRTI groups in TA and plasma (using the age-unadjusted analyses), only 15 proteins were differentially expressed in both compartments, with seven proteins upregulated in vLRTI in both, four proteins downregulated in vLRTI in both, and four proteins with opposite directionality (Fig. 3D). We further investigated protein correlation utilizing our paired samples ($n = 48$ total paired TA and plasma samples from the same subject, including $n = 30$ paired vLRTI samples and $n = 18$ paired No LRTI samples). Correlation in expression between the lower airway and systemic circulation was weak for the majority of proteins (Pearson correlation coefficient −0.3 to + 0.3) (Fig. 3E), although there were exceptions, namely ISG-15.

## Lower respiratory tract proteomic differences in bacterial-viral coinfection

Within the vLRTI group, subjects were categorized as either viral infection ($n = 16$) or bacterial-viral coinfection ($n = 24$) based on clinical microbiology and respiratory mNGS. Differential protein expression in TA between these two groups did not yield any statistically significant proteins at $p_{adj}$ <0.05, but we did note an absolute increase in the expression of TSG-6, a tumor-necrosis factor-stimulated protein ($p_{adj} = 0.07$), and C-reactive protein (CRP) ($p_{adj} = 0.10$) in coinfection (Fig. 4A). Pathway analysis showed heightened interferon signaling in coinfection compared with viral infection alone.

**TABLE 1** Demographic and clinical characteristics of the vLRTI and No LRTI cohorts

|  | vLRTI (*n* = 40) | No LRTI (*n* = 22) | *P* value[a] |
|---|---|---|---|
| Female, n (%) | 18 (45.0%) | 11 (50.0%) | 0.79 |
| Age in years, median (IQR) | 0.9 (0.3–1.6) | 10.2 (1.1–14.9) | <0.01 |
| Race |  |  | 0.71 |
| White, n (%) | 26 (65.0%) | 14 (63.6%) |  |
| Black/African American, n (%) | 6 (15.0%) | 6 (27.3%) |  |
| Asian, n (%) | 1 (2.5%) | 1 (4.5%) |  |
| Native Hawaiian/Pacific Islander, n (%) | 1 (2.5%) | 0 (0.0%) |  |
| American Indian/Alaska Native, n (%) | 1 (2.5%) | 0 (0.0%) |  |
| Unknown/not reported, n (%) | 5 (12.5%) | 1 (4.5%) |  |
| Hispanic/Latino ethnicity, n (%) | 8 (20.0%) | 1 (4.5%) | 0.14 |
| Comorbidity, n (%) | 14 (35.0%) | 11 (50.0%) | 0.29 |
| Immunocompromise, n (%) | 1 (2.5%) | 0 (0.0%) | 0.99 |
| Admission category |  |  | <0.01 |
| Medical, n (%) | 40 (100%) | 14 (63.6%) |  |
| Surgical, n (%) | 0 (0.0%) | 5 (22.7%) |  |
| Trauma, n (%) | 0 (0.0%) | 3 (13.6%) |  |
| Infiltrates on initial CXR, n (%) | 36 (90.0%) | 12 (54.5%) | <0.01 |
| ARDS, n (%) | 18 (45.0%) | 4 (18.0%) | 0.05 |
| Received antibiotics, n (%) | 14 (35.0%) | 8 (36.4%) | 0.99 |
| Ventilator days, median (IQR) | 6 (5.0–9.0) | 6 (5.0–8.0) | 0.85 |
| ICU length of stay in days, median (IQR) | 10 (8.0–16.5) | 9 (6.3–13.3) | 0.25 |
| Hospital length of stay in day, median (IQR) | 14 (10.5–19.5) | 16 (9.5–38.8) | 0.34 |
| Mortality, n (%) | 1 (2.5%) | 3 (13.6%) | 0.12 |

[a]Wilcoxon rank sum test used for all continuous variables. Fisher's exact test is used for all categorical variables. IQR, interquartile range; ARDS, acute respiratory distress syndrome; ICU, intensive care unit.

Pathways associated with cell turnover and division were preferentially upregulated in viral infection compared with coinfection (Fig. 4B).

### Lower respiratory tract protein correlations with viral load

For the vLRTI subjects that tested positive for a virus by mNGS, the expression of TA proteins was correlated with viral load, measured as reads-per-million (Fig. 5). Interferon-related proteins, such as interferon-lambda 1 and ISG-15, were positively correlated with viral load, as well as monocyte chemotactic protein MCP-2. Conversely, platelet receptor GI-24, TFG-β superfamily protein Activin AB, and neutrophil-activating glycoprotein CD177 were inversely correlated with viral load.

### DISCUSSION

In this study, we identified the proteomic signature of severe pediatric vLRTI in both the lower respiratory tract and systemic circulation, leveraging results to understand compartment-specific host responses, host-viral dynamics, and viral-bacterial coinfection, as well as identify specific proteins with diagnostic potential. This work represents the first simultaneous proteomic profiling of both TA and plasma samples from children with severe vLRTI.

As hypothesized, the proteomic response to vLRTI was most robust at the local site of infection, with approximately 15% of assayed proteins differentially expressed in TA. This lower airway proteomic vLRTI signature was dominated by interferon-related proteins, which are well-known innate mediators of host defense and immunologic injury in viral infection (23), and importantly also shown *in vitro* to be structurally altered during certain viral infections (24, 25). In addition, this signature was enriched in proteins contained in cytotoxic lymphocytes that in turn secrete interferons (26). The list of upregulated proteins and pathways share significant overlap with a smaller study that performed proteomic profiling on nasal swab samples from adults with influenza,

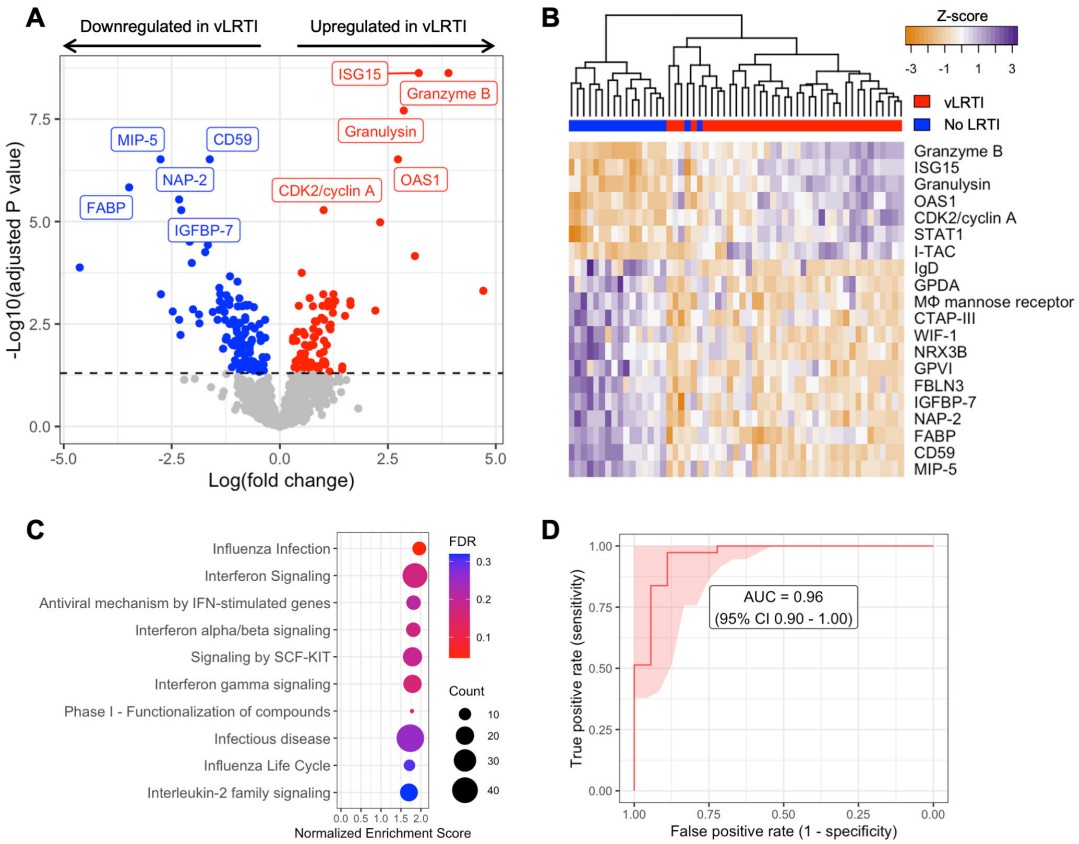

**FIG 2** Comparison of host protein expression between vLRTI and No LRTI cohorts in tracheal aspirate. (A) Volcano plot of the differentially expressed proteins, with proteins significantly upregulated in vLRTI in red, and proteins significantly downregulated in vLRTI in blue. The top 10 proteins based on $P_{adj}$ are labeled. (B) Heat map showing differential expression of the top 20 proteins based on $P_{adj}$ (rows) across all patients (columns). Dendrogram clustering (top) highlights the proteomic differences between the two groups. (C) Pathway analysis showing the top 10 pathways (all upregulated) ordered by Normalized Enrichment Score. Dot color indicates the false discovery rate (FDR) $P_{adj}$, and size indicates the number of proteins included in the pathway. (D) Receiver operator characteristic (ROC) curve of the proteomic classifier to distinguish vLRTI from No LRTI.

suggesting overlapping immune responses across the upper and lower respiratory tract (27). Notable downregulated proteins were macrophage inhibitory protein-5 (MIP-5) and fatty acid binding protein (FABP), which have a diverse array of functions including macrophage regulation, suggesting that macrophage dynamics play an important role in response to vLRTI (28, 29). Interestingly, in the subanalysis of bacterial-viral coinfection, the expression of interferon-related proteins was even greater than in viral infection alone. Prior work, mostly in influenza infections, has suggested that type 1 interferons can suppress key neutrophil and macrophage defenses, increasing susceptibility to bacterial coinfection, which may explain this finding (30, 31).

By integrating viral load measurements, we identified host TA proteins exhibiting proportional changes in expression based on viral load. Interferon-related proteins, including ISG-15 and interferon-lambda 1, and MCP-2, a chemokine-induced by interferon signaling, exhibited the strongest induction in expression with viral load, underscoring the central role of interferons in innate antiviral defense. In contrast, the levels of CD177 (a glycoprotein involved in neutrophil activation) (32), Activin AB (a TGF-β family protein implicated in ARDS inflammatory remodeling) (33), and GI-24 (a platelet aggregation receptor) (34) all decreased in response to higher viral loads. As previously noted, impaired neutrophil responses have been implicated in the pathophysiology of post-viral bacterial pneumonia (30, 31), and our results suggest that this may occur in a viral load-dependent manner. Complementing these findings, a longitudinal transcriptomic study in adults hospitalized with severe influenza infection demonstrated the

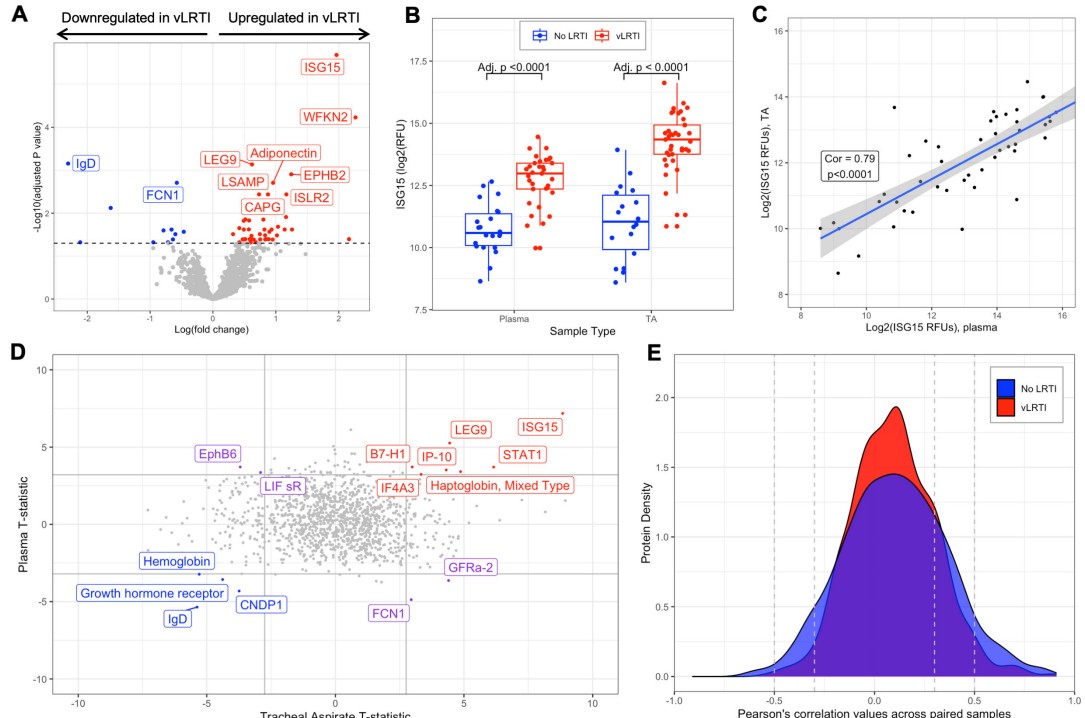

**FIG 3** Host-protein expression in plasma and comparative proteomic analysis between plasma and tracheal aspirate samples. (A) Volcano plot of the differentially expressed plasma proteins in vLRTI versus No LRTI (age unadjusted). The top 10 proteins based on $P_{adj}$ are labeled. (B) Ubiquitin-like ISG-15 protein expression in plasma (left) and tracheal aspirate (TA) (right) in vLRTI (red) and No LRTI (blue) groups. (C) Log-log plot showing the correlation of ISG-15 values between paired plasma (x-axis) and TA (y-axis) samples. The shaded region represents the 95% CI for the correlation. (D) T statistics for each protein calculated with limma for vLRTI versus No LRTI comparisons in plasma and TA were plotted against one another. Proteins highlighted in red were significantly upregulated across both body compartments, those highlighted in blue were downregulated in both, and those highlighted in purple deviated in opposite directions. (E) Density plot showing correlation coefficients for each protein in TA versus plasma, with stratification based on group (vLRTI in red vs No LRTI in blue).

initial upregulation of interferon pathways, followed by inflammatory neutrophil activation and cell-stress patterns (35), and a study of severe pediatric influenza infection found that early upregulation of genes associated with neutrophil degranulation was associated with multi-organ dysfunction and mortality (36).

Although we observed a robust protein signature of vLRTI in the lower airways, the findings in the peripheral blood were more subtle, and the correlation between plasma and TA proteins was generally weak. Furthermore, we observed a greater impact of age on the blood proteomic signature of vLRTI, potentially because the signal in the peripheral blood was weaker and thus more susceptible to confounding. Understanding the systemic response to a local infection is certainly useful and practical, as peripheral blood samples and urine samples are less invasive to collect than lower respiratory samples and would allow for application in a broader population of children who do not require MV. However, to obtain the most informative and potent proteomic signal of infection, our findings suggest that sampling the site of infection has the highest yield. Supporting this intuitive finding is a comparative adult proteomic study assaying both serum and bronchoalveolar lavage in interstitial lung diseases that similarly found a much higher number of differentially expressed proteins in the lower respiratory tract compared with the blood (37).

Increasingly, high-dimensional proteomic assays including mass spectrometry and antibody-based methods like SomaScan have been employed in different sample types to identify novel biomarkers for a wide range of disease states (38–41). In addition to contributing insights into the pathophysiology of vLRTI, our study also highlights the utility of proteomic approaches in diagnostic biomarker discovery specific to respiratory infections. Standard-of-care multiplexed PCR assays only evaluate a limited subset

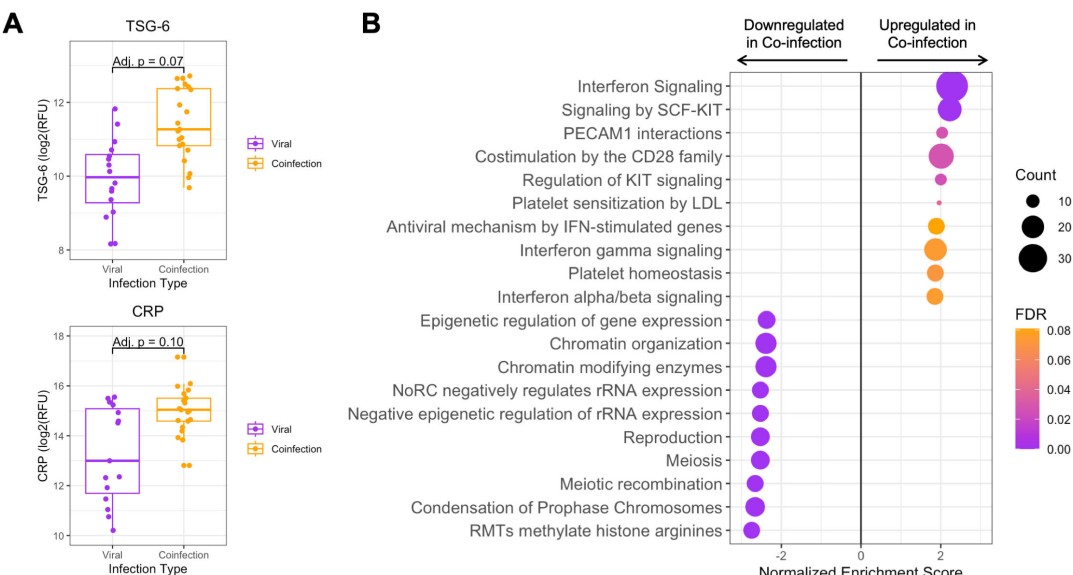

**FIG 4** Tracheal aspirate protein and pathway expression in bacterial-viral coinfection. (A) Box plots of the two most differentially expressed proteins, tumor necrosis factor-stimulated gene-6 (TSG-6), and C-reactive protein (CRP) between viral infection and coinfection subgroups. (B) Pathway analysis showing the top 20 pathways up- or down-regulated in bacterial-viral coinfection compared with viral infection alone. Dot color indicates the false discovery rate (FDR) $P_{adj}$, and the size indicates the number of proteins included in the pathway.

of respiratory viruses (42) and cannot detect novel emerging viruses or differentiate asymptomatic carriage from true infection (43). Host response-based assays agnostic to viral species could be invaluable for pandemic preparedness and infection prevention in congregate settings. When employed as a diagnostic test to distinguish vLRTI from non-infectious respiratory failure, our nine-protein TA classifier achieved excellent performance with an AUC of 0.96. The single protein ISG15 also showed potential for use as a diagnostic biomarker in both TA and plasma. Type 1 interferons have previously been proposed as an accurate diagnostic screening test for pediatric viral infection (44). Another diagnostic challenge in vLRTI is identifying bacterial coinfection, as standard respiratory bacterial cultures do not distinguish between coinfection and colonization and are often negative in the context of prior antibiotic administration. Our subanalysis of bacterial coinfection highlighted two TA proteins, TSG-6 (a tumor necrosis factor-inducible protein) and CRP (an inflammatory protein with modest specificity for bacterial LRTI in blood) (45), that may be useful respiratory biomarkers of secondary bacterial infection.

Our study has several strengths including our multi-center enrollment, clinical sampling at early time points, evaluation of protein expression in multiple compartments, and integration of respiratory mNGS for comprehensive pathogen evaluation. It also has several important limitations, including a small sample size, which may have limited our ability to detect more subtle but clinically important differences in protein expression. Additionally, the version of the SomaScan assay utilized does not encompass the entire human proteome, and we likely missed some important differentially expressed proteins and pathways. Evaluation of a larger number of proteins using mass spectrometry, OLINK, or the newer SomaScan 11 k would enable a more comprehensive evaluation of the proteome (46, 47). From the diagnostic biomarker standpoint, our findings are more preliminary in nature and warrant further optimization and validation in larger cohorts with all relevant classes of infection (bacterial infection, viral infection, coinfection, and non-infectious controls) and a wider range of severity represented to rigorously understand performance. Finally, we recognize that there is no gold standard for LRTI diagnosis in children, and our reliance on the best practical methodology of combining retrospective clinical adjudication and microbiology results may have resulted in classification errors.

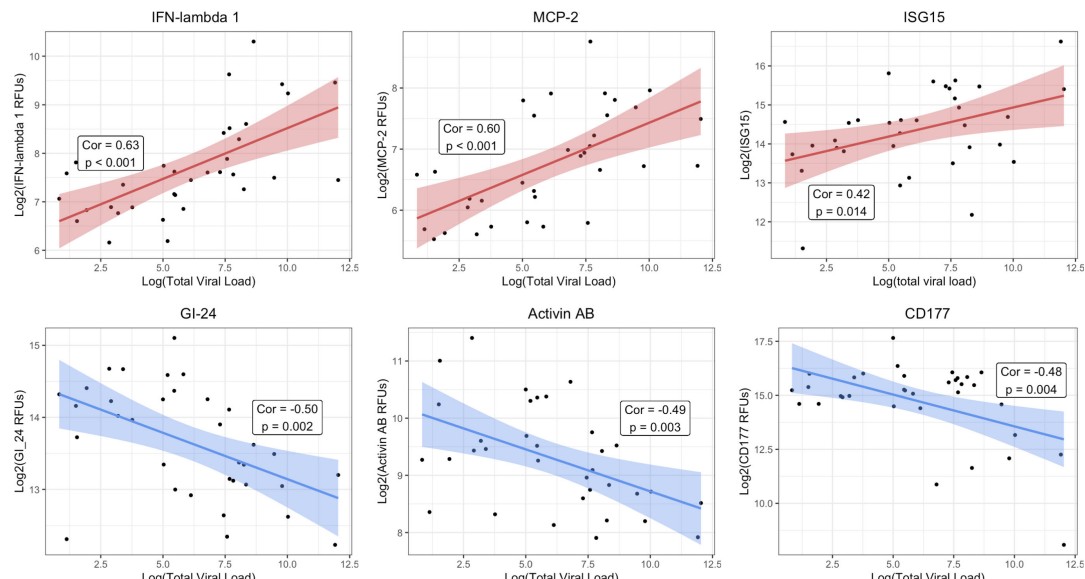

**FIG 5** Correlation of lower respiratory protein expression and viral load. For each vLRTI subject with a virus detected on mNGS, correlation coefficients were calculated between viral load and relative concentration of each protein in TA. The top three highest positive correlations and the top three negative correlations are shown: interferon lambda-1 (IFN-lambda 1), monocyte chemotactic protein-2 (MCP-2); ubiquitin-like interferon-stimulated gene-15 (ISG-15), platelet receptor GI-24, activin AB, and CD177. The shaded region represents the 95% CI for each correlation.

Taken together, we present a comprehensive proteomic characterization of severe pediatric vLRTI, highlighting pathophysiologic insights in both viral infection and bacterial-viral coinfection and deepening our understanding of compartmentalization of the human host response to LRTI. Validation of the present findings in larger external cohorts is needed with more in-depth analysis to determine whether new therapeutic targets can be identified and whether proteomic biomarkers may augment current standard-of-care pathogen-based diagnostic testing. Looking forward, multi-omic approaches combining proteomics and transcriptomics as well as integration with microbiology hold promise for advancing understanding of the heterogeneity of pediatric LRTI, modernizing diagnostics, and personalizing treatment.

## ACKNOWLEDGMENTS

Support was provided by the Eunice Kennedy Shriver National Institute of Child Health and Human Development, the National Heart, Lung, and Blood Institute: UG1HD083171 and 1R01HL124103 (Dr. Mourani), UG1HD049983 (Dr. Carcillo), UG1HD083170 (Dr. Hall), UG1HD050096 (Dr. Meert), UG1HD63108 (Dr. Zuppa), UG1HD083116 (Dr. Sapru), UG1HD083166 (Dr. McQuillen), UG1HD049981 (Dr. Pollack), K23HL138461, and 5R01HL155418 (Dr. Langelier). The study was also supported by funding from the Chan Zuckerberg Biohub. The study sponsors were not involved in study design; in the collection, analysis, and interpretation of data; in the writing of the report; and in the decision to submit the report for publication.

## AUTHOR AFFILIATIONS

[1]Department of Medicine, University of California San Francisco, San Francisco, California, USA

[2]Department of Anesthesiology and Critical Care Medicine, Children's Hospital of Philadelphia, Philadelphia, Pennsylvania, USA

[3]Department of Pediatrics, Children's Hospital of Philadelphia, Philadelphia, Pennsylvania, USA

⁴Department of Biostatistics and Informatics, University of Colorado, Colorado School of Public Health, Aurora, Colorado, USA

⁵Sections of Emergency Medicine and Hospital Medicine, Children's Hospital Colorado, Aurora, Colorado, USA

⁶Department of Pediatrics, University of Colorado School of Medicine and Children's Hospital Colorado, Aurora, Colorado, USA

⁷Department of Pediatrics, University of Utah, Salt Lake City, Utah, USA

⁸Chan Zuckerberg Biohub, San Francisco, California, USA

⁹Department of Biochemistry and Biophysics, University of California San Francisco, San Francisco, California, USA

¹⁰Department of Pediatrics, Nationwide Children's Hospital, Columbus, Ohio, USA

¹¹Departments of Pediatrics and Critical Care Medicine, University of Pittsburgh, Pittsburgh, Pennsylvania, USA

¹²Department of Pediatrics, Children's Hospital of Michigan, Central Michigan University, Detroit, Michigan, USA

¹³Department of Pediatrics, University of California Los Angeles, Los Angeles, California, USA

¹⁴Department of Pediatrics,, Children's National Medical Center and George Washington School of Medicine and Health Sciences, Washington, DC, USA

¹⁵Department of Pediatrics, University of California San Francisco, San Francisco, California, USA

¹⁶Department of Molecular Biology, Princeton University, Princeton, New Jersey, USA

¹⁷Department of Pediatrics, Critical Care, University of Arkansas for Medical Sciences and Arkansas Children's Hospital, Little Rock, Arkansas, USA

## AUTHOR ORCIDs

Emily Lydon  http://orcid.org/0000-0003-3510-6254
Christina M. Osborne  http://orcid.org/0000-0002-7452-4734
Charles R. Langelier  http://orcid.org/0000-0002-6708-4646
Peter M. Mourani  http://orcid.org/0000-0002-1829-3775

## FUNDING

| Funder | Grant(s) | Author(s) |
| --- | --- | --- |
| HHS | NIH | National Heart, Lung, and Blood Institute (NHLBI) | 1R01HL124103 | Peter M. Mourani |
| HHS | NIH | National Heart, Lung, and Blood Institute (NHLBI) | 5R01HL155418 | Charles R. Langelier |
| | | Peter M. Mourani |
| HHS | NIH | National Heart, Lung, and Blood Institute (NHLBI) | UG1HD083171 | Peter M. Mourani |
| Chan Zuckerberg Biohub | | Joseph L. DeRisi |

## AUTHOR CONTRIBUTIONS

Emily Lydon, Conceptualization, Data curation, Formal analysis, Investigation, Methodology, Validation, Visualization, Writing – original draft, Writing – review and editing | Christina M. Osborne, Formal analysis, Investigation, Writing – original draft, Writing – review and editing | Brandie D. Wagner, Formal analysis, Investigation, Methodology, Supervision, Writing – original draft, Writing – review and editing | Lilliam Ambroggio, Investigation, Project administration, Supervision, Writing – review and editing | J. Kirk Harris, Investigation, Methodology, Writing – review and editing | Ron Reeder, Data curation, Writing – review and editing | Todd C. Carpenter, Methodology, Writing – review and editing | Aline B. Maddux, Data curation, Investigation, Writing – review and editing | Matthew K. Leroue, Writing – review and editing | Nadir Yehya, Writing – review and editing | Joseph L. DeRisi, Methodology, Writing – review and editing | Mark W. Hall,

Writing – review and editing | Athena F. Zuppa, Writing – review and editing | Kathleen Meert, Writing – review and editing | Anil Sapru, Writing – review and editing | Murray M. Pollack, Writing – review and editing | Patrick McQuillen, Writing – review and editing.

## DATA AVAILABILITY

Proteomic data, subject metadata, and code for reproducing the results of this study can be found at: https://github.com/infectiousdisease-langelier-lab/pedsLRTIproteomics.

## ADDITIONAL FILES

The following material is available online.

### Supplemental Material

**Supplemental Material (mSystems01335-24-s0001.docx).** Supplemental methods, table, and figures.

### Open Peer Review

**PEER REVIEW HISTORY (review-history.pdf).** An accounting of the reviewer comments and feedback.

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
