## [Reviewer comments · mSystems]

Proteomic profiling of the local and systemic immune response to pediatric respiratory viral infections

Emily Lydon, Christina Osborne, Brandie Wagner, Lilliam Ambroggio, Jonathan Harris, Ron Reeder, Todd Carpenter, Aline Maddux, Matthew Leroue, Nadir Yehya, Joseph DeRisi, Mark Hall, Athena Zuppa, Joseph Carcillo, Kathleen Meert, Anil Sapru, Murray Pollack, Patrick McQuillen, Daniel Notterman, Charles Langelier, and Peter Mourani

Corresponding Author(s): Charles Langelier, University of California San Francisco

Review Timeline:

Submission Date:	October 7, 2024
Editorial Decision:	November 5, 2024
Revision Received:	November 9, 2024
Accepted:	November 11, 2024

Editor: Neha Garg

Reviewer(s): The reviewers have opted to remain anonymous.

Transaction Report:

DOI: <https://doi.org/10.1128/msystems.01335-24>

Re: mSystems01335-24 (Proteomic profiling of the local and systemic immune response to pediatric respiratory viral infections)

Dear Dr. Charles R Langelier:

Thank you for the privilege of reviewing your work. Below you will find instructions from the mSystems editorial office, and the reviewer comments, which are minor and addressed through textual modifications.

Please return the manuscript within 30 days; if you cannot complete the modification within this time period, please contact me. If you do not wish to modify the manuscript and prefer to submit it to another journal, notify me immediately so that the manuscript may be formally withdrawn from consideration by mSystems.

Revision Guidelines

Sincerely,
Neha Garg
Editor
mSystems

Reviewer #1 (Comments for the Author):

In this manuscript, the authors describe using SomaScan proteomics of tracheal aspirate and plasma from pediatric viral lower respiratory tract infection vs non-infectious acute respiratory failure subjects. The subject is important. The manuscript is clear and well written. The experimental methods and data analysis are good. I have only a single minor issue.

Minor issue:

Figures 3C, 5: The shaded region is not defined. It should be defined or removed.

Reviewer #2 (Comments for the Author):

This work describes proteomic analyses of pediatric respiratory viral infections and identifies potential diagnostic biomarkers. The study is concise, well designed and well explained. The results support the conclusions and the authors have identified strengths and limitations of their study.

My only comments for suggested improvements relate to the Discussion. The authors state (lines 288-289) that their study "highlights the utility of proteomic approaches in diagnostic biomarker study". While true, many other previous studies also did. These include: PMIDs: 26906940 and 27528862, and several that used the same aptamer platform (ie. PMIDs: 27051355 and 28452194). In addition, PMID: 27088501 also used SomaScan to investigate respiratory infections. It would be worthwhile to acknowledge some of these prior studies and to compare/contrast results. Finally, a large number of in vitro proteomic studies also have investigated respiratory virus-induced proteomic alterations, and many identified ISG15, OAS1, and similar molecules to those identified in this study. While acknowledging these are in vitro rather than in vivo, and while an exhaustive review of these other studies is not needed, some comparisons and general conclusions are warranted.

The authors acknowledge (lines 310-311) that the SomaScan version they used only interrogated ~ 1300 proteins. A brief description of some of the newer technologies that can interrogate many more proteins, including newer SomaScan versions, is warranted.

November 6, 2024

Dear Dr. Garg:

We greatly appreciate your and the reviewers' thoughtful reviews and comments on our manuscript entitled, "Proteomic profiling of the local and systemic immune response to pediatric respiratory viral infections" (mSystems01335-24). We have carefully considered each comment and suggestion and have directly addressed each one of them. We believe that the feedback has strengthened the manuscript.

Below we provide the point-by-point responses raised by each reviewer. All modifications in the manuscript have been tracked in a marked-up file, and a clean version has been submitted as well. As instructed, we have removed the figures from the main manuscript file and have attached them separately as high-resolution TIFF files.

Thank you for your consideration,

Charles Langelier, MD, PhD
Associate Professor of Medicine
Division of Infectious Diseases
University of California, San Francisco
chaz.langelier@ucsf.edu

Reviewer #1's comments:

In this manuscript, the authors describe using SomaScan proteomics of tracheal aspirate and plasma from pediatric viral lower respiratory tract infection vs non-infectious acute respiratory failure subjects. The subject is important. The manuscript is clear and well written. The experimental methods and data analysis are good. I have only a single minor issue.

C1: Minor issue: Figures 3C, 5: The shaded region is not defined. It should be defined or removed.

R1: We thank the reviewer for this comment. The shaded region in each graph represents the 95% confidence interval for the correlation. We have added this detail to the figure captions for Figure 3c and Figure 5.

Reviewer #2's comments:

This work describes proteomic analyses of pediatric respiratory viral infections and identifies potential diagnostic biomarkers. The study is concise, well designed and well explained. The results support the conclusions and the authors have identified strengths and limitations of their study.

C2: My only comments for suggested improvements relate to the Discussion. The authors state (lines 288-289) that their study "highlights the utility of proteomic approaches in diagnostic biomarker study". While true, many other previous studies also did. These include: PMIDs: 26906940 and 27528862, and several that used the same aptamer platform (ie. PMIDs: 27051355 and 28452194). In addition, PMID: 27088501 also used SomaScan to investigate respiratory infections. It would be worthwhile to acknowledge some of these prior studies and to compare/contrast results. Finally, a large number of in vitro proteomic studies also have investigated respiratory virus-induced proteomic alterations, and many identified ISG15, OAS1, and similar molecules to those identified in this study. While acknowledging these are in vitro rather than in vivo, and while an exhaustive review of these other studies is not needed, some comparisons and general conclusions are warranted.

R2: We read with interest the articles suggested by the reviewer. We have added each as a citation, highlighting the important preceding work using proteomics for biomarker discovery in non-infectious conditions (PMIDs 26906940, 27528862, 27051355, and 28452194) with the following text:

Line 241-245: "Increasingly, high-dimensional proteomic assays including mass spectrometry and antibody-based methods like SomaScan have been employed in different sample types to identify novel biomarkers for a wide range of disease states.(38-41) In addition to contributing insights into the pathophysiology of vLRTI, our study also highlights the utility of proteomic approaches in diagnostic biomarker discovery specific to respiratory infections."

Earlier in the discussion, we now also highlight the study that performed proteomic profiling in influenza infections using SomaScan (PMID 27088501) which showed a significant amount of overlap with our results:

Line 201-204: "The list of upregulated proteins and pathways share significant overlap with a smaller study that performed proteomic profiling on nasal swab samples from adults with influenza, suggesting overlapping immune responses across the upper and

lower respiratory tract.(27)”

We also added two citations (PMID 34663977 and 34342578) highlighting *in vitro* alterations of key interferon-related proteins that were shown to be upregulated in our study:

Line 197-200: “This lower airway proteomic vLRTI signature was dominated by interferon-related proteins, which are well-known innate mediators of host defense and immunologic injury in viral infection,(23) and importantly also shown *in vitro* to be structurally altered during certain viral infections.(24-25)”

C3: The authors acknowledge (lines 310-311) that the SomaScan version they used only interrogated ~ 1300 proteins. A brief description of some of the newer technologies that can interrogate many more proteins, including newer SomaScan versions, is warranted.

R3: We agree with the reviewer that explicitly mentioning the newest version of SomaScan and other higher-dimensional proteomic assays is important. We have also added two additional sources that cover these technologies in more detail (PMID 39038188 and 35598103):

Line 267-269: “Evaluation of a larger number of proteins using mass spectrometry, OLINK, or the newer SomaScan® 11k would enable more comprehensive evaluation of the proteome.(46-47)

Re: mSystems01335-24R1 (Proteomic profiling of the local and systemic immune response to pediatric respiratory viral infections)

Dear Dr. Charles R Langelier:

Your manuscript has been accepted, and I am forwarding it to the ASM production staff for publication. Your paper will first be checked to make sure all elements meet the technical requirements. ASM staff will contact you if anything needs to be revised before copyediting and production can begin. Otherwise, you will be notified when your proofs are ready to be viewed.

Sincerely,
Neha Garg
Editor
mSystems